# Real-World Evidence on Palliative Gemcitabine and Oxaliplatin (GemOx) Combination Chemotherapy in Advanced Biliary Tract Cancer

**DOI:** 10.3390/cancers13143507

**Published:** 2021-07-13

**Authors:** Hanna Lagenfelt, Hakon Blomstrand, Nils O. Elander

**Affiliations:** 1Department of Clinical and Biomedical Sciences, Linköping University, 58183 Linköping, Sweden; hanna.lagenfelt@regionostergotland.se (H.L.); hakon.andersen.blomstrand@regionostergotland.se (H.B.); 2Department of Oncology, Linköping University, 58183 Linköping, Sweden; 3Department of Clinical Pathology, Linköping University, 58183 Linköping, Sweden

**Keywords:** real world, biliary tract neoplasms, gemcitabine, oxaliplatin, palliative care, chemotherapy

## Abstract

**Simple Summary:**

Cancers of the biliary tract are rare but severe with high mortality rates. Randomised controlled trials suggest that chemotherapies such as gemcitabine and oxaliplatin (GemOx) may relieve symptoms and prolong life, but less is known on the efficacy and safety of such regimens in real life. The current paper assessed the real-world outcome of GemOx in all patients with advanced biliary tract cancer treated at any cancer centre in the South East Region of Sweden over a period of nine years. The median overall survival was nine months and time to disease progression five months. Prognostic factors such as performance status and gall bladder (rather than bile duct) localisation of the primary tumour were identified. Most patients received a lower dose of oxaliplatin than proposed by previous studies, which seemed feasible as few patients had severe adverse events. This study supports further use of GemOx as standard of care.

**Abstract:**

Background: Gemcitabine and oxaliplatin (GemOx) is a standard combination regimen in advanced biliary tract cancer (BTC). There is limited evidence on its efficacy and safety in real life. Methods: A retrospective multicentre cohort study in the South East Region of Sweden, covering nine years (2011–2020) and three hospitals where GemOx was treatment of choice, was designed. Clinicopathological prognostic parameters were explored. Results: One hundred and twenty-one patients with advanced BTC were identified. Median overall and progression-free survival (OS and PFS) were 8.9 (95% CI = 7.2–10.6) and 5.3 (95% CI = 3.8–6.7) months. Performance status according to Eastern Cooperative Oncology Group (PS according to ECOG) 1–2 and primary gallbladder carcinoma were independent predictors for poor OS. PS and derived neutrophil/lymphocyte ratio were predictive for PFS. The most common severe type of myelosuppresion was grade 3 neutropenia that was recorded in 8%. Fifty-three (43.8%) experienced at least one episode of unplanned hospitalisation. One hundred and seventeen (97%) received oxaliplatin with lower dosage than was utilized in previous phase III trials (80–85 vs. 100 mg/m^2^) and a majority received further dose reductions of oxaliplatin and/or gemcitabine. Conclusion: The outcome of GemOx in advanced BTC appears comparable in controlled trials and real-world contexts. A lower dose of oxaliplatin seems more tolerable without compromising the outcome.

## 1. Introduction

Biliary tract cancer (BTC) is an entity comprising a group of rare types of cancers with high mortality rates, including intrahepatic cholangiocarcinoma, perihilar cholangiocarcinoma (Klatskin tumour), extrahepatic cholangiocarcinoma (distal biliary tract carcinoma) and gallbladder carcinoma [1]. In general, the incidence of cholangiocarcinoma varies between 0.3 and 6 cases per 100,000, but in China, South Korea and Thailand, incidence rates of >7 are seen [2]. For gallbladder carcinoma, the worldwide age standardized incidence rate (per 100,000) is 0.9 for men and 1.4 for women [3]. The median 5-year survival varies between 10 and 40%, which is highly dependent on the location of the primary tumour, the disease stage at diagnosis and access/eligibility for multimodal treatment in terms of surgery and/or chemotherapy [4,5].

The majority of patients with BTC have locally advanced and/or metastatic disease at the time of diagnosis, both of which preclude curative intent surgery. Over the last twenty years, gemcitabine (Gem) has been a cornerstone in palliative systemic treatment of advanced BTC, either alone or in combination with platinum compounds [6,7]. The additional value of the gemcitabine/cisplatin (GemCis) combination (compared to Gem monotherapy) was confirmed in the milestone phase III trial ABC-02 by Valle et al. from 2010 [8], which displayed an improvement of almost four months in median overall survival (OS) in the combination arm (11.7 vs. 8.1 months for GemCis and Gem, respectively). Not surprisingly, the improved survival with GemCis came at the expense of increased toxicity. To avoid adverse events and complications specifically associated with cisplatin (e.g., ototoxicity, acute kidney failure and nausea/vomiting), alternative combinations with Gem and other platinum compounds such as oxaliplatin (GemOx) have been introduced in parallel.

The efficacies of the two separate regimes GemCis and GemOx were described in a systematic review [9]. While OS appeared to be marginally better with the GemCis combination (11.7 vs. 9.7 months for GemCis and GemOx, respectively), toxicity profile and tolerability rather favoured the GemOx regimen. On the other hand, a recent phase III trial on advanced gallbladder cancer failed to show superiority for either of the two regimens, although OS was numerically slightly better in the GemOx arm [10]. A Cochrane review on standard BTC treatments such as GemCis and GemOx concluded that there was little evidence for superiority of either of the two regimens [11].

Due to the favourable toxicity profile, many centres, including all oncology departments in the South East Region of Sweden, have advocated GemOx (rather than GemCis) as standard first-line treatment in advanced BTC. So far, there is limited published evidence on the feasibility, efficacy and safety of the GemOx regimen in the real-life setting. As the health care in Sweden is publicly funded and available for all citizens regardless of socioeconomic status, the conditions for evaluating real-world outcomes are optimal. A population-based multicentre retrospective cohort study was therefore designed to assess the real-world outcome and safety profile of GemOx in advanced BTC, covering all eligible patients in the geographical area over a period of nine years. Besides providing real-world data on overall and progression-free survival (OS and PFS), potentially prognostic parameters in terms of clinical, pathological and biochemical characteristics were explored. In addition, key palliative parameters such as access to specialised palliative care and chemotherapy at end of life were analysed.

## 2. Materials and Methods

### 2.1. Patients

A retrospective multicentre cohort study was conducted in the South East Region of Sweden, including the oncology departments of Linköping, Jönköping and Kalmar. The area has a population of approximately 1.1 million citizens. All included patients were identified using the digital software CSAM Cytodose (CSAM Health AS, Oslo, Norway), which was the software used for prescribing chemotherapy at all participating centres. Inclusion criteria were as follows: administration of at least one dose of palliative first-line GemOx for biliary tract carcinoma (intrahepatic, perihilar and extrahepatic/distal cholangiocarcinoma) or gallbladder carcinoma (ICD codes C23.x, C24.x and C22.1) at any of the participating centres between November 2011 and September 2020. Patients who only received GemOx in the neoadjuvant or adjuvant setting were excluded. Otherwise, and to reflect the real-world situation, no exclusion criteria were applied.

### 2.2. Treatment

The GemOx chemotherapy regimens used were slightly different at the participating sites and over the study period. While gemcitabine was consistently prescribed in 1000 mg/m^2^ at day one, the oxaliplatin dose was either 80, 85 mg/m^2^ or (in a very limited number of patients) 100 mg/m^2^ and was given either at day one or day two every 14-day cycle. If oxaliplatin was omitted due to toxicity, subsequent gemcitabine monotherapy cycles were still considered part of the first-line GemOx regimen. To equalize the registration of the total number of cycles, any 28-day cycles in which gemcitabine was prescribed following the omission of oxaliplatin were registered as two 14-day cycles.

Slight progression after a planned treatment break and restart of the same regimen did not count as true progression. If treatment was initiated with dose reduction merely for ‘tolerance testing’, but then escalated to full dose within two cycles and the patient received more than one cycle of full dose, the higher dose was considered the starting dose.

### 2.3. Patient Data

Clinical data were collected from medical records with a structured case report form and included baseline patient data, tumour characteristics, history of previous neoadjuvant/adjuvant chemotherapy and curative intent surgery, treatment intensity and duration, toxicity, blood samples at baseline, haematological toxicity (graded according to Common Terminology Criteria for Adverse Events (CTCAE) version 5.0), unplanned hospitalisations, admission to specialist palliative care and any further chemotherapy treatment after GemOx. All patients were followed until death or until 31 December 2020, whatever came first. As lymphocyte counts were not routinely analysed, derived neutrophil lymphocyte ratio (dNLR) [12] was utilized as a surrogate for NLR, and was calculated according to the formula below.
$$\frac{\mathrm{neutrophils}}{\mathrm{white}\;\mathrm{blood}\;\mathrm{cell}\;\mathrm{count}-\mathrm{neutrophils}}$$

The cutoff value used was adopted from Grenader et al. and has been shown to predict OS and PFS in advanced BTC [13].

### 2.4. Statistics

Statistical analyses were performed using SPSS Statistics v25 (IBM, Armonk, NY, USA). Primary outcome was overall survival (OS), counted from start of treatment until death or last follow-up date. Secondary outcomes included progression-free survival (PFS), counted from start of treatment until progression. Progression was determined from radiology reports or as determined by the managing clinician. Other relevant outcomes were unplanned hospitalisations, infections requiring antibiotics and/or hospitalisation, haematological toxicity, peripheral neurotoxicity, number of treatment cycles, dose intensity, treatment within last 30 days of life and inclusion into specialist palliative care. Median OS and PFS were estimated using Kaplan–Meier survival analysis and the significance of the difference between factors was calculated using Mantel–Cox log rank test. To evaluate hazard ratios for potential prognostic factors, Cox regression analysis was performed. *p*-values < 0.05 were considered significant. Statistically significant prognostic factors in the univariate analysis were further analysed in a multivariate Cox regression analysis.

### 2.5. Ethics

Ethical approval for this study was granted by the Regional Ethics Review board in Linköping (diary number 2018/139-31). Due to the retrospective non-interventional design and the fact that the majority of patients were not expected to be alive at time of data collection, the Ethics board waived the requirement for informed consent.

## 3. Results

### 3.1. Patients

Over the study period (2011–2020), a total of 171 patients treated with at least one cycle of GemOx, at any of the covered oncology centres in the South East Region of Sweden, were identified. Fifty of these were excluded due to exclusion criteria, leaving a total cohort of 121 patients receiving first-line palliative intent GemOx for advanced BTC (Figure 1). Patient and treatment characteristics are displayed in Table 1. A minority of patients (*n* = 35, 29%) had previously been treated with curative intent resection, and 17 (14%) had received previous adjuvant chemotherapy, corresponding to approximately half of the patients (*n* = 17, 49%) that underwent surgery. The most commonly used adjuvant chemotherapy was capecitabine (*n* = 11, 65%). Most patients had one organ with metastasis at date of incurable disease (*n* = 59, 49%). The most common site of metastasis was the liver (*n* = 36, 30%) followed by peritoneum (*n* = 27, 22%) and lymph nodes (*n* = 24, 20%). A majority of patients (*n* = 101, 84%) had the diagnosis verified with histology or cytology.

### 3.2. Treatment Tolerability and Toxicity

At baseline (i.e., before the first cycle of palliative GemOx), most patients were performance status 1 (*n* = 56, 49%) followed by 0 (*n* = 40, 35%) and 2 (*n* = 19, 17%). GemOx was administrated with gemcitabine at 1000 mg/m^2^ in all patients (100%), whereas oxaliplatin was dosed at 100 mg/m^2^ in 4 patients (3%), 85 mg/m^2^ in 66 patients (55%) and 80 mg/m^2^ in 51 patients (42%). About two thirds of patients received full dose of gemcitabine (*n* = 82, 68%) and full dose of oxaliplatin (*n*= 81, 67%) at treatment start. The main reason for initial dose reduction was general health deterioration followed by elevated liver enzymes. Dose reduction later on was performed in 91 (75%) patients regarding oxaliplatin and 68 (56%) patients regarding gemcitabine. The main reason for dose reduction at this stage was peripheral neurotoxicity (*n* = 33, 27%), followed by haematological toxicity (*n* = 32, 26%).

Median number of treatment cycles was six if counting only combination therapy and seven if counting total number of treatment cycles, including subsequent gemcitabine monotherapy following discontinuation of oxaliplatin. The main reason for termination of treatment was progression (*n* = 72, 60%) and impaired performance status (*n* = 18, 15%). Sixty-two (53%) patients received additional line(s) of chemotherapy after GemOx (Table 1).

At least one episode of infection requiring antibiotics and/or hospitalisation was recorded in 51 (42%) patients during the GemOx treatment period. Three cases (2.5%) of febrile neutropenia were observed. Peripheral neurotoxicity affecting activities of daily life (corresponding to chemotherapy-induced peripheral neurotoxicity grades 3–4 according to the CTCAE scale) was recorded in 33 (27%) of patients. Fifty-three (44%) patients had at least one episode of unplanned (inward) hospitalisation. Grade 3 haematotoxicity was evident in five patients (4.2%) regarding anaemia, four (3.4%) regarding thrombocytopenia and leukopenia and six (8.2%) regarding neutropenia (Table 2). No cases of grade 4 myelosuppression were observed.

### 3.3. Survival

The median OS for the whole cohort was 8.9 months (95% CI 7.2–10.6) and median PFS was 5.3 months (95% CI 3.8–6.7). The median follow-up time was 8.7 months and at end of follow up, 116 patients had evidence of progressive disease (95%) and 108 patients had died (89%).

### 3.4. Univariate Analyses

Regression analyses and Kaplan–Meier survival graphs on overall survival are shown in Table 3 and Figure 2. Patients with ECOG performance status (PS) of 1–2 had significantly shorter OS compared to patients with performance status 0 (HR 2.3, 95% CI 1.4–3.6, *p* = 0.001, corresponding to OS 12.5 months (95% CI 8.9–16.1) and 8.2 (95% CI 7.1–10.7), respectively). Other parameters for poor prognosis were female sex (HR = 1.6, 95% CI 1.1–2.4, *p* = 0.017), serum albumin < 35 g/L (HR = 1.5, 95% CI 1.1–2.2, *p* = 0.03), primary tumour location of the gallbladder (HR 2.3, 95% CI 1.4—3.5, *p* < 0.001) and high dNLR (HR 1.8, 95% CI 1.1–3.0, *p* = 0.031). No lymph node metastases at primary diagnosis was associated with better prognosis (HR 0.6, 95% CI 0.4–0.9, *p* = 0.016).

Similar results were observed when the same analyses were applied on PFS, as shown in Table 4 and Figure 3, although baseline serum albumin and lymph node metastases at primary diagnosis did not reach statistical significance.

The other parameters analysed, i.e., previous curative intent surgery performed, age </≥ 65, serum ALP at baseline >/≤ 2, metastatic burden, previous adjuvant chemotherapy, lung metastases at primary diagnosis, liver metastases at primary diagnosis or peritoneal metastases at primary diagnosis, were neither statistically significant with regard to OS nor PFS.

### 3.5. Multivariate Analysis

In multivariate analysis regarding OS, including performance status, gender, albumin, dNLR, lymph nodes metastases at primary diagnosis and gallbladder cancer, high performance status and gallbladder cancer remained independent risk factors for worse OS (HR 2.6, 95% CI 1.5–4.6, *p* = 0.001 and HR 1.9, 95% CI 1.0–3.5, *p* = 0.036, Table 3).

On PFS, corresponding multivariate analyses revealed that performance status and high dNLR were independent prognostic factors (HR 2.1, 95% CI 1.2–3.5, *p* = 0.007 and HR 2.4, 95% CI 1.4–4.3, *p* = 0.002, Table 4).

### 3.6. Treatment in End of Life and Palliative Care Admissions

Median number of days from last dose of gemcitabine until death was 123.5 days (range 2–1183) in the total cohort. When excluding patients who received other treatments beyond progression on GemOx, the median number of days was 61.5 (range 2–455). In this latter subgroup of patients, 10 (19%) received their last cycle of GemOx within the last 30 days of life.

Seventy-four (61%) of the patients were admitted to a specialised hospital-based palliative care team. The median number of days from diagnosis of incurable disease until inclusion in palliative care was 260.5 (range 7–1491). Median number of days from inclusion in palliative care until death was 36 (range 0–1059, Table 5).

## 4. Discussion

To our knowledge, this is the first published multicentre real-world cohort study on first-line GemOx combination chemotherapy in advanced BTC. Median OS and PFS were 8.9 and 5.3 months, respectively. While the outcomes closely mirror what was previously reported in the phase III trial by Sharma et al. [10], which compared GemOx to GemCis in patients with gallbladder cancer and reported mOS of 9 and 8.3 months in the GemOx and GemCis arms, respectively, the outcomes appear slightly worse than the corresponding results of the ‘Gemcitabine and oxaliplatin with or without cetuximab in advanced biliary-tract cancer’ (BINGO) trial [14], where survival in the GemOx comparator arm was 12.4 months (vs. 11 months in the experimental GemOx/cetuximab arm). This minor difference is not surprising, as the BINGO trial only enrolled patients with PS 0–1, whereas the present cohort included PS 2 as well.

Notably, a similarly conducted real-world study on Taiwanese patients [15], but treated with GemCis, reported almost identical data to ours.

In the present study, patients with gallbladder cancer had a significantly worse OS than patients with primary tumour in intra- or extrahepatic bile ducts. Previous studies have shown diverse results regarding tumour site, where some studies are in line with ours [16,17] while others rather suggest intrahepatic tumours to be particularly bad, (e.g., Andre et al., [7,18]). The milestone ABC-02 trial, however, reported no prognostic impact of the primary tumour site [8].

Female gender was associated with poor OS in univariate analysis but did not reach significance in the multivariate analysis. Notably, there was a slight male predominance in all diagnoses but gallbladder cancer, which was in contrast characterised by a marked female overrepresentation. This is in line with previous data that have shown a female to male ratio of 3–6:1 [19,20]. This sex difference in OS could therefore likely be explained by the presence of gallbladder cancer and not by gender per se.

Not surprisingly, poor performance status was associated with both worse OS and shorter PFS. This has been shown in multiple previous studies [18,21,22,23], which emphasise the poor prognosis in frail patients and add doubts to the value of palliative chemotherapy in patients with impaired performance status.

The present study reports low incidence of grade 3–4 bone marrow toxicity. While no patient experienced grade 4 toxicity, only 8% of patients had the most common type of myelosuppression in terms of grade 3 neutropenia. The incidence of grade 3 anaemia and thrombocytopenia was even lower. These figures are generally lower than observed in other studies. In the study by Andre et al., 14% of the population experienced grade 3–4 neutropenia, whereas 9% had grade 3–4 thrombocytopenia and/or anaemia. The BINGO trial [14] reported grade 4 neutropenia in 3% and grade 3 in 13% of the patients in the GemOx comparator arm. Thrombocytopenia grade 3 was recorded in 19% in the same group but with no grade 4 toxicity.

It is reasonable to believe that the less frequent and less severe bone marrow toxicity observed in the present cohort may depend on the more cautious dosing of oxaliplatin. In our study, only four patients received oxaliplatin 100 mg/m^2^ (that was the standard dosing in the Andre and BINGO trials), while the vast majority (117 out of 121) received oxaliplatin at 80 mg/m^2^ or 85 mg/m^2^. In addition, about one third of patients received upfront dose reduction of gemcitabine and/or oxaliplatin. Given the closely resembling survival estimates in the different studies, it therefore seems that a slightly lower starting dose of oxaliplatin is more feasible in terms of tolerance but is similarly potent as the higher dosage utilised in earlier trials.

Apart from myelosuppression, chemotherapy-induced peripheral neuropathy (CIPN) is a common complication for platinum compounds. In the present cohort, 27% of patients experienced CIPN with negative impact on activities of daily life and, following myelosuppression, it was the second most common reason for dose reduction during the treatment course. In 31% of patients, oxaliplatin was, at some point, discontinued completely and replaced with the single drug gemcitabine. However, while the main reason for ultimate termination of the treatment was progression followed by impaired performance status, toxicity was the reason to terminate in just a few (6%) of the patients.

There was quite a high proportion (44%) of patients with at least one episode of unplanned inward hospitalisation in the present population. Forty-two percent of the patients had at least one infection that required antibiotic use and/or hospitalisation. As febrile neutropenia was very rare, it appears that these events were related to the disease rather than the chemotherapy per se, although the retrospective nature of the study and the lack of an untreated control group preclude firm conclusions on the matter. Nevertheless, this underlines the frailty of these types of patients and implies that the treating oncologist should be alert for signs of infection and/or general deterioration of the performance status.

The medical centres covered by this study did not routinely analyse whole differential white blood cell counts, making it impossible to perform calculations on NLR, which previously have been shown to be a prognostic marker for both OS and PFS [24,25,26]. Some studies have however investigated the use of dNLR and found it to be a reasonable surrogate for NLR [13]. As dNLR builds on the absolute neutrophil count and the total number of white blood cells, it is possible to estimate even without lymphocyte counts available. Notably, dNLR was found to be a significant prognostic factor for both OS and PFS in univariate analyses of the present population. Following multivariate analysis, the statistical significance remained for PFS but not OS. This is still in line with a previous study by Grenader et al. [13], who reported dNLR to be a strong independent prognostic factor for patients with advanced BTC in the compiled populations of the ABC-02 [8] and one additional study [27] in terms of both OS and PFS.

For those patients that did not receive further lines of treatment after GemOx, more than 80% did not receive any chemotherapy in the last 30 days of life. Notably, this is a somewhat higher proportion than was reported in a study by Randén et al. [28] that focused on near end-of-life chemotherapy in patients with advanced cancer in Stockholm. This indicates that few patients were ‘over-treated’ with GemOx in the end-of-life situation.

While 63% of the patients received specialised hospital-based palliative care, the median number of days from inclusion into specialised palliative care to death was limited to 36 days. Although these data are difficult to interpret, as ‘simpler’ forms of palliative care might have been provided by other care givers not covered by the present application, such as general practitioners, it still indicates that early admission to a palliative care provider is recommended as the disease course may rapidly accelerate in these types of patients. Clinical experience as well as evidence-based guidelines [29,30] suggest that access to a skilled palliative care provider is essential for preserving quality of life in late stage cancer.

The present study has some essential limitations. There was no standard regimen or overall guidance on starting doses or protocol-based dose reductions, although the vast majority received a lower dose of oxaliplatin than was initially stipulated in clinical trials. Still, we believe it is of considerable interest to evaluate the outcome and safety in an unselected patient cohort receiving treatment according to the oncologist’s discretion rather than according to a strict protocol. To account for the low incidence of this disease, the inclusion period covered a long period of time (nine years) in order to achieve a reasonably large cohort.

The most obvious strength is the true real-world approach, with a population-based inclusion covering all patients, regardless of socioeconomic status, who received this treatment in a large geographical area. While formal assessment of quality of life was not possible given the retrospective design, significant efforts were undertaken to collect detailed data on toxicity, unplanned hospitalisations and admissions to palliative care providers, which may all be considered surrogates for symptom burden and quality of life.

Together with other clinical trials and real-world studies on the topic, it is obvious that novel and more potent treatment strategies are needed in order to obtain considerably improved survival and disease control in advanced BTC.

## 5. Conclusions

This study provides novel and robust real-world data on GemOx combination chemotherapy in advanced BTC. Survival estimates are similar to the outcome in comparable clinical trials, which supports further utilisation of GemOx in the palliative treatment of BTC. Lower starting doses of oxaliplatin (80–85 mg/m^2^ rather than 100 mg/m^2^) seem to be associated with lower frequency of severe bone marrow toxicity without apparently impairing the prognosis. Good performance status is a strong and independent prognostic factor, both in terms of OS and PFS. In addition, primary tumour location in the gallbladder and dNLR were recognized as independent parameters predictive for OS and PFS, respectively. Frequent infections and unplanned hospitalisations, as well as the short expected survival in these types of patients, indicate that early admission to a specialised palliative care provider is advisable. Novel treatment strategies are necessary to improve the long-term prognosis in advanced BTC.

## Figures and Tables

**Figure 1 cancers-13-03507-f001:**
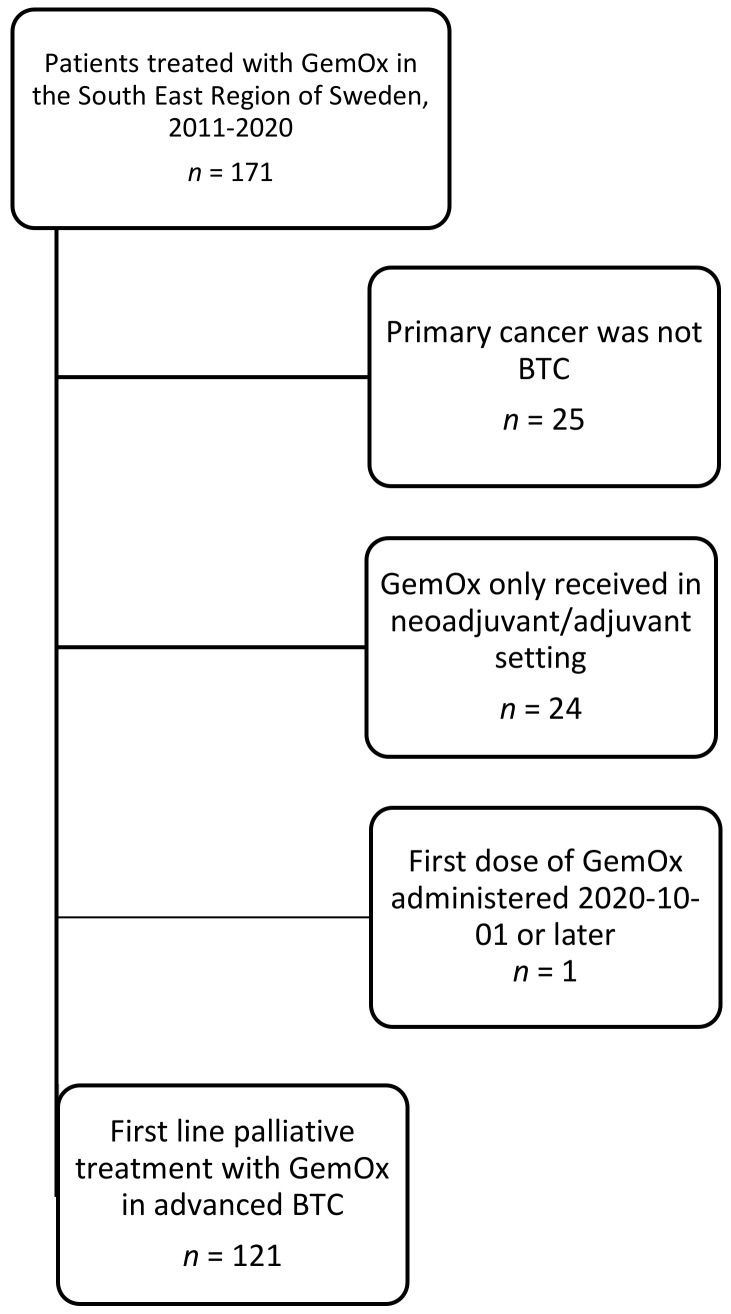
Flow chart of study population. BTC: biliary tract cancer; GemOx: gemcitabine plus oxaliplatin; *n*: number of patients.

**Figure 2 cancers-13-03507-f002:**
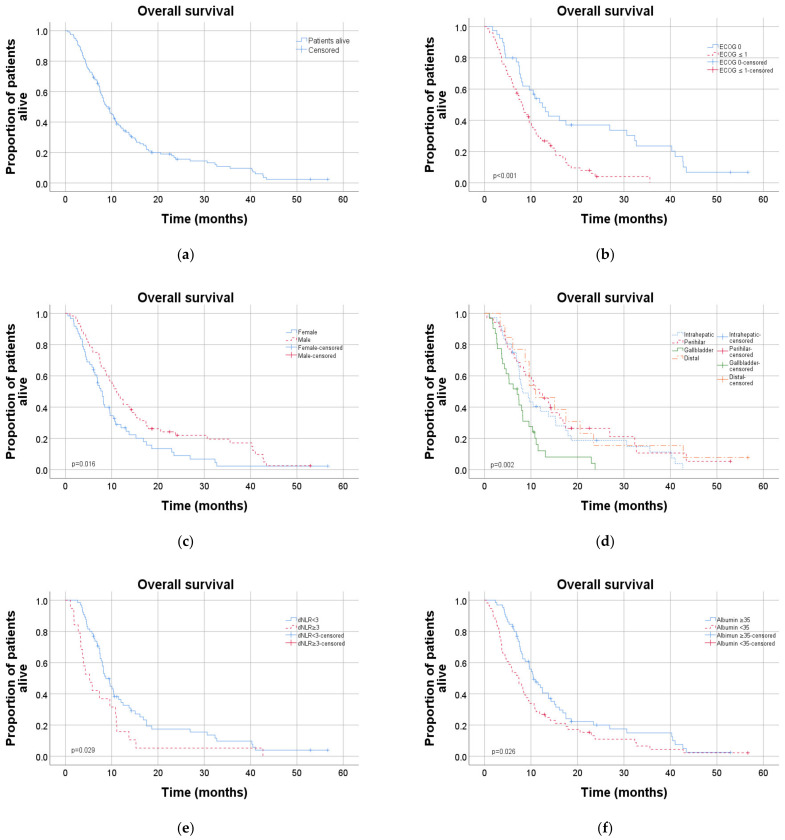
Kaplan–Meier diagrams showing overall survival for different subgroups: (**a**) total cohort, (**b**) performance status, (**c**) gender, (**d**) diagnosis, (**e**) dNLR at baseline and (**f**) serum albumin at baseline. dNLR: derived neutrophil/lymphocyte ratio.

**Figure 3 cancers-13-03507-f003:**
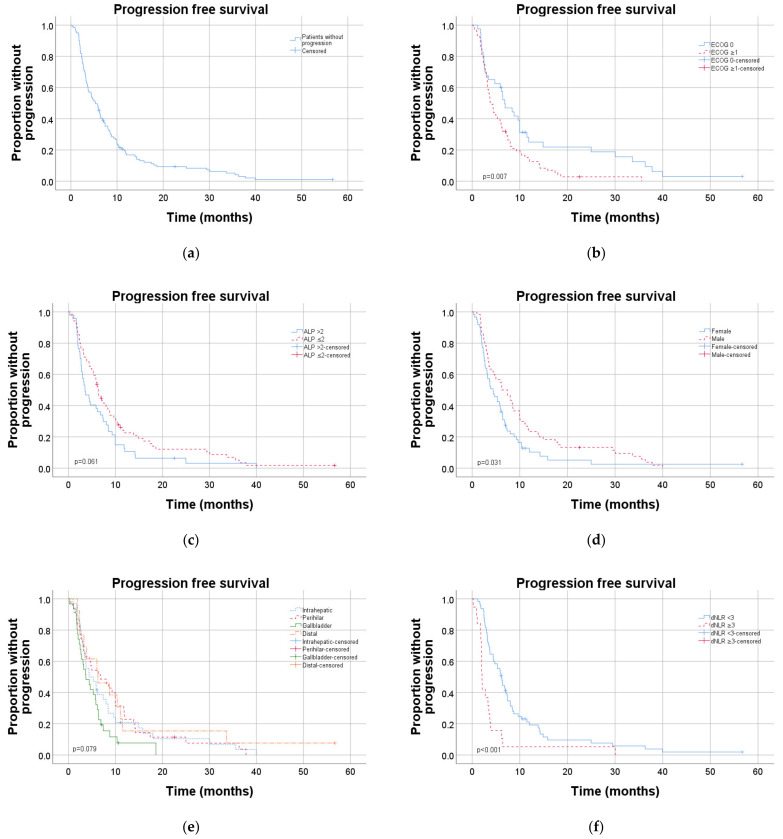
Kaplan–Meier diagrams showing progression-free survival for different subgroups: (**a**) total cohort, (**b**) performance status, (**c**) ALP at baseline, (**d**) gender, (**e**) diagnosis and (**f**) dNLR at baseline. dNLR: derived neutrophil/lymphocyte ratio.

**Table 1 cancers-13-03507-t001:** Patient, tumour and treatment characteristics.

Patient, Tumour and Treatment Characteristics	Number (%)
**Gender**	
Male	60 (50.4)
Female	61 (49.6)
**Age**	
≥65	60 (49.6)
<65	61 (50.4)
**ECOG performance status**	
0	40 (34.8)
1	56 (48.7)
2	19 (16.5)
**Tumour stage**	
Locally advanced	31 (25.6)
Metastasised	72 (59.5)
**Number of metastases at baseline**	
0	26 (21.5)
1	59 (48.8)
2	25 (20.7)
>2	11 (9.1)
**Metastatic site(s) in patients with upfront metastasised disease**	
Liver	36 (29.8)
Peritoneum	27 (22.3)
Lymph nodes	24 (19.8)
Lung	16 (13.2)
Other	7 (5.8)
**Metastatic site(s) in patients with recurrent disease**	
Liver	17 (48.6)
Local relapse	16 (45.7)
Lymph nodes	9 (25.7)
Lung	5 (14.3)
Peritoneum	4 (11.4)
Other	1 (2.9)
**Previous chemotherapy**	
Adjuvant	17 (14)
Neoadjuvant	3 (2.5)
**Drugs included in adjuvant chemotherapy**	
Capecitabine	11 (64.7)
Gemcitabine	2 (11.8)
Gemcitabine/Oxaliplatin	2 (11.8)
Gemcitabine/Capecitabine	1 (5.9)
5-FU/Capecitabine	1 (5.9)
**Histology/cytology verified diagnosis**	
Yes	101 (83.5)
No	20 (16.5)
**Previous curative intent surgery performed**	
Yes	35 (28.9)
No	86 (71.1)
**Site of primary tumour**	
Intrahepatic	36 (30.0)
Perihilar	35 (29.0)
Gallbladder	31 (25.6)
Distal	13 (10.7)
Missing data	6 (5.0)
**Plasma albumin at baseline**	
<35	56 (46.3)
≥35	65 (53.7)
**Full dose at start**	
Gemcitabine	82 (67.8)
Oxaliplatin	81 (66.9)
**Reason for dose reduction at start**	
General health deterioration	11 (9.1)
Elevated liver enzymes	8 (6.6)
Adverse event from previous adjuvant therapy	6 (5)
Age	4 (3.3)
Other	5 (4.1)
Unknown reason	12 (9.9)
**Dose reduction later on**	
Gemcitabine	68 (56.2)
Oxaliplatin	91 (75.2)
**Reason for dose reduction later on**	
Peripheral neurotoxicity	33 (27.3)
Haematological	32 (26.4)
Elevated liver enzymes	10 (8.3)
Gastrointestinal	1 (0.8)
Infection	1 (0.8)
Renal	0 (0)
Other	17 (14.0)
**Discontinuation of oxaliplatin before discontinuation of gemcitabine**	
Yes	38 (31.4)
**Treatment after GemOx**	
Yes	62 (51.2)
No Unknown/censored	54 (44.6) 5 (4.1)
**Drugs covered by second and later lines of treatment after GemOx**	
Capecitabine	55 (88.7)
5-FU	17 (27.4)
Irinotecan	9 (14.5)
Cisplatin	1 (1.6)
Carboplatin	1 (1.6)
Other	6 (9.7)
**Main reason for termination of GemOx**	
Progression	72 (59.5)
Impaired performance status	18 (14.9)
Toxicity	7 (5.8)
Elevated liver enzymes	6 (5.0)
Stable disease/planned treatment holiday	5 (4.1)
Death	5 (4.1)
Ongoing treatment	2 (1.7)
Other	6 (5.0)
**Number of treatment cycles GemOx, median (range)**	6 (1–21)
**Number of treatment cycles Gem, median (range)**	7 (1–69)
**Alkaline phosphatase (ALP)**	
>2	47 (40.5)
≤2	69 (59.5)
**dNLR**	
<3.0	65 (77.4)
≥3.0	19 (22.6)

**Table 2 cancers-13-03507-t002:** Presence of bone marrow toxicity according to CTCAE, *n* (%).

Type of Bone Marrow Toxicity	0	1	2	3	4
Anaemia	20 (16.8)	67 (56.3)	27 (22.7)	5 (4.2)	0 (0)
Thrombocytopenia	30 (25.2)	73 (61.3)	12 (10.1)	4 (3.4)	0 (0)
Leukopenia	94 (79.0)	5 (4.2)	16 (13.4)	4 (3.4)	0 (0)
Neutropenia	47 (64.4)	5 (6.8)	15 (20.5)	6 (8.2)	0 (0)

**Table 3 cancers-13-03507-t003:** Table showing median overall survival for different factors with univariate and multivariate analysis.

Patient and Tumour Characteristics	OS (95% CI)	HR Univariate (95% CI)	*p*	HR Multivariate (95% CI)	*p*
Total cohort	**8.9 (7.2–10.6)**				
Performance status				2.6 (1.5–4.6)	**0.001**
ECOG 0	12.5 (8.9–16.1)				
ECOG > 0	8.2 (7.1–10.7)	2.3 (1.4–3.6)	**0.001**		
Gender					
Male	11.0 (8.8–13.3)				
Female	8.0 (6.6–9.3)	1.6 (1.1–2.4)	**0.017**	1.0 (0.6–1.7)	0.89
Curative intent surgery performed					
Yes	8.0 (6.6–9.3)				
No	9.6 (7.4–11.8)	1.0 (0.7–1.6)	0.883		
Age					
≥65	8.2 (5.9–10.5)				
<65	9.7 (7.5–11.8)	0.9 (0.6–1.3)	0.605		
Albumin					
≥35	10.5 (8.0–13.0)				
<35	7.4 (4.7–10.0)	1.5 (1.1–2.2)	**0.03**	1.4 (0.8–2.4)	0.241
ALP					
>2	8.2 (6.8–9.6)				
≤2	10.3 (8.2–12.5)	0.7 (0.5–1.1)	0.096		
Metastatic burden					
0–1	10.1 (7.9–12.3)				
≥2	7.8 (6.9–8.6)	1.4 (0.9–2.2)	0.091		
Previous adjuvant chemotherapy					
Yes	7.5 (5.8–9.3)				
No	9.7 (7.7–11.6)	0.7 (0.4–1.2)	0.156		
dNLR					
<3.0	8.7 (7.2–10.3)				
≥3.0	5.2 (2.5–7.9)	1.8 (1.1–3.0)	**0.031**	1.4 (0.8–2.6)	0.235
Gallbladder cancer					**0.036**
No	10.4 (8.1–12.8)				
Yes	7.0 (4.2–9.9)	2.3 (1.4–3.5)	**<0.001**	1.9 (1.0–3.5)	
Lymph nodes by primary diagnosis					0.624
Yes	6.8 (2.9–10.8)				
No	10.1 (7.6–12.5)	0.6 (0.4–0.9)	**0.016**	0.9 (0.5–1.6)	
Lung metastases by primary diagnosis					
Yes	7.8 (2.9-12.6)				
No	8.9 (7.1-10.7)	0.9 (0.5-1.6)	0.728		
Peritoneum metastases by primary diagnosis					
Yes	10.1 (4.8–15.4)				
No	8.9 (7.5–10.2)	0.9 (0.6–1.4)	0.622		
Liver metastases by primary diagnosis					
Yes	7.8 (4.7–10.8)				
No	9.4 (7.4–11.5)	0.8 (0.5–1.2)	0.325		

**Table 4 cancers-13-03507-t004:** Table showing median progression-free survival for different factors with univariate and multivariate analysis.

Patient and Tumour Characteristics	PFS (95% CI)	HR Univariate (95% CI)	*p*	HR Multivariate (95% CI)	*p*
**Total cohort**	**5.3 (3.8–6.7)**				
**Performance status**					
ECOG 0	6.8 (4.1–9.5)				
ECOG > 0	3.9 (3.0–4.7)	1.8 (1.2–2.7)	**0.008**	2.1 (1.2–3.5)	**0.007**
Gender					
Male	6.3 (3.6–9.0)				
Female	4.4 (2.8–6.0)	1.5 (1.0–2.2)	**0.032**	1.0 (0.6–1.6)	0.893
**Curative intent surgery performed**					
Yes	5.7 (3.8–7.7)				
No	5.3 (3.2–7.4)	1.1 (0.7–1.7)	0.636		
**Age**					
≥65	3.9 (1.8–6.0)				
<65	6.1 (4.1–8.2)	0.9 (0.6–1.3)	0.705		
**Albumin**					
≥35	6.3 (5.1–7.4)				
<35	3.7 (2.7–4.8)	1.3 (0.9–1.8)	0.217		
**ALP**					
>2	3.5 (2.0–5.1)				
≤2	6.4 (5.2–7.5)	0.7 (0.5–1.0)	0.064		
**Metastatic burden**					
0–1	6.1 (4.9–7.4)				
≥2	3.9 (2.3–5.5)	1.3 (0.9–2.0)	0.188		
**Previous adjuvant chemotherapy**					
Yes	3.4 (1.2–5.6)				
No	5.7 (4.2–7.3)	0.6 (0.4–1.0)	0.063		
**dNLR**					
<3.0	6.1 (4.9–7.4)				
≥3.0	2.1 (1.9–2.2)	2.7 (1.6–4.6)	**<0.001**	2.4 (1.4–4.3)	**0.002**
**Gallbladder cancer**		1.7 (1.1–2.7)	**0.015**	1.4 (0.8–2.4)	0.249
No	5.9 (3.8–8.0)				
Yes	3.7 (2.0–5.4)				
**Lymph nodes by primary diagnosis**		0.8 (0.5–1.2)	0.259		
Yes	3.7 (2.1–5.4)				
No	5.9 (4.4–7.4)				
**Lung metastases by primary diagnosis**		0.7 (0.4–1.1)	0.14		
Yes	3.3 (2.35–4.28)				
No	5.9 (4.7–7.2)				
**Peritoneum metastasis by primary diagnosis**		0.7 (0.5–1.1)	0.16		
Yes	3.7 (1.8–5.7)				
No	5.7 (4.1–7.4)				
**Liver metastases by primary diagnosis**		0.8 (0.6–1.2)	0.325		
Yes	3.6 (2.7–4.6)				
No	6.1 (4.5–7.7)				

**Table 5 cancers-13-03507-t005:** Table showing inclusion into specialist palliative care and near end-of-life treatment.

Parameter	Number (%) or Median (Range)
Inclusion into specialist palliative care, *n* (%)	74 (61.2)
Days from incurable diagnosis until inclusion in palliative care, median (range)	260.5 (7–1491)
Days from inclusion in palliative care until death, median (range)	36 (0–1059)
Days from last dose of gemcitabine until death, median (range)	
Total	123.5 (2–1183)
Treatment after Gem	166.5 (37–1183)
No treatment after Gem	61.5 (2–455)
Patients receiving treatment with GemOx in the last 30 days of life	
Total	10 (8.3)
No treatment after GemOx	10 (18.5)

## Data Availability

Additional data are available at reasonable request to the corresponding author.

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
