# Peer review of "Real-World Evidence on Palliative Gemcitabine and Oxaliplatin (GemOx) Combination Chemotherapy in Advanced Biliary Tract Cancer"

_cancers, 2021, doi:10.3390/cancers13143507_

Round 1

Reviewer 1 Report

This study provides real-world data on the GemOX regimen for patients with BTC. The results show a survival benefit comparable to that of prospective clinical trials, indicating that the GemOX regimen is versatile and useful chemotherapy. Although it is a retrospective study, it is a valuable report and deserves to be published. Therefore, we request the authors to do the following.

According to the results of previous clinical trials, the notable toxicity of the GemOX regimen is “neuropathy". It is important to consider whether this has affected the patient's quality of life on patients. The table that shows the toxicity profile only shows myelosuppression. I would recommend that the authors describe the frequency and grade of neuropathy associated with GemOX chemotherapy.

This is an editing problem, but in Table 1, there are a few places where the line for the item and the number of patients (%) are out of place.

Finally, I think this research is very meaningful.

Author Response

We are very grateful for the constructive review and the encouraging comments.

Please find the requested information about peripheral neuropathy at page 7 (lines 181 – 183). Regarding myelosuppression, it was possible to extract the exact values and corresponding CTCAE grades from the medical records. As this was a retrospective study, no formal CTCAE scoring of subjective symptoms (such as peripheral neuropathy) was evident from the medical records. The data on neuropathy with impact on activities of daily life were therefore extracted from ‘free text’ descriptions in the medical records. Due to the lack of formal scoring of peripheral neuropathy, we did not consider it suitable to include it in the same Table as myelosuppression.

The Tables look correct in my computer, but probably something goes wrong when they are viewed in another context. I presume that the editorial team will resolve this – if not, please let me know what to do.

Reviewer 2 Report

While the subject is of certain interest, the present study escape key points in Cholangiocarcinoma. 

  1. Authors did include all types of cholangiocarcinoma along with gallbladar carcinoma in one cohort. The study of each type of cancer separately should be addressed.
  2. The study does not add information or possibile new treatment for cholangiocarcinoma, as cholangiocarcinoma is characterised with extremely high mortality, high recurrence and high cheoressitance including to GemOX therapy, new strategy are absolutly needed. 
  3. In methods authors mentioned that Patients who only received GemOx in the neoadjuvant or adjuvant setting were excluded. So how could they conclude results about GemOX therapy while it was only considered in comination with other treatment strategies. 
  4. The study is good statestical retrospective description, but it did not provide a new prospective on diagnostic, prognostic or therapeutic level to cholangiocarcinoma. 

Author Response

  1. We are very grateful for the constructive review including valuable comments and questions.

    1. Authors did include all types of cholangiocarcinoma along with gallbladar carcinoma in one cohort. The study of each type of cancer separately should be addressed. Reply: As all types of biliary tract cancers are rare, they are often treated similarly in clinical trials and guidelines. The distribution of the various subtypes is shown in Table 1. To address whether there were any differences between gall bladder cancer and cholangiocarcinoma, Kaplan Meier survival curves including estimation of median overall and progression free survival, as well as univariate and multivariate Cox regression analyses, were performed to compare the different subtypes. Please find these data in Tables 3 and 4 and Figures 2 and 3, as well as throughout the Results section.
  2. The study does not add information or possibile new treatment for cholangiocarcinoma, as cholangiocarcinoma is characterised with extremely high mortality, high recurrence and high cheoressitance including to GemOX therapy, new strategy are absolutly needed. Reply: This study aimed to describe the clinical outcome of GemOx in a long term follow up real world cohort. Novel and/or experimental treatments which, self-evidently, are not offered in standard practice, were beyond the scope of this study. The need for future novel treatment strategies has been highlighted in the last sentences of the Discussion and Conclusion sections of the manuscript (page 14; lines 354-356 and lines 368-369).
  3. In methods authors mentioned that Patients who only received GemOx in the neoadjuvant or adjuvant setting were excluded. So how could they conclude results about GemOX therapy while it was only considered in comination with other treatment strategies. Reply: We apologise if we have not understood this question correctly. The patients in this retrospective cohort study were identified through the chemotherapy prescription software (Cytodose). Following careful review of the individual medical records, it was evident that some patients identified had only received GemOx in the neoadjuvant or adjuvant (i.e. curative intent) setting. As this study aimed to assess the outcome of GemOx in the palliative setting, these patients were therefore excluded. Though, patients who had undergone resection with or without neoadjuvant/adjuvant chemotherapy and, later on, had been diagnosed with recurrent/metastatic disease and treated with GemOx in the palliative setting, were all included.
  4. The study is good statestical retrospective description, but it did not provide a new prospective on diagnostic, prognostic or therapeutic level to cholangiocarcinoma. Reply: The scope of this study was to provide real world evidence on GemOx in patients with advanced BTC, i.e. to describe the outcome and safety of this established regimen in patients treated outside the frames of a prospective trial. The need for novel improved treatment strategies have been highlighted in Discussion and Conclusion sections (page 14; lines 354-356 and lines 368-369).

Reviewer 3 Report

The authors investigated the efficacy on palliative Gemcitabine and Oxaliplatin Combination Chemotherapy in Advanced Biliary tract cancer.

They refered in detail the tolerability of Gemcitabine and Oxaliplatin Combination Chemotherapy for Gemcitabine and Cisplatin Combination Chemotherapy.

Therefore, I will be able to recommend it for acceptance with several points.

# It would be better to mention whether there are any other drugs that are considered equally or more useful than Gemcitabine and Oxaliplatin Combination Chemotherapy (GemOx), and whether a comparative study of GemOx with those drugs should be considered.

Author Response

We are very grateful for the constructive review and valuable comments.

To improve the quality of the language, the manuscript has now undergone a careful proofreading by a colleague who is a native Englishman and very experienced in scientific writing.

Regarding the need for novel and more potent drugs in BTC, this has been highlighted in the last paragraph of the Discussion (page 14; lines 354-356) and the last sentence of the Conclusion section (page 14; lines 368-369).

Round 2

Reviewer 2 Report

The authors have responded adequately to the questions.